# Emotions, Stress and Coping among Healthcare Workers in a Reproductive Medicine Unit during the First and Second COVID-19 Lockdowns

**DOI:** 10.3390/ijerph19105899

**Published:** 2022-05-12

**Authors:** Marcella Paterlini, Erica Neri, Alessia Nicoli, Federica Genova, Maria Teresa Villani, Sara Santi, Francesca Agostini

**Affiliations:** 1Department of Obstetrics and Pediatrics, AUSL-IRCCS, 42123 Reggio Emilia, Italy; marcella.paterlini@ausl.re.it; 2Department of Psychology “Renzo Canestrari”, University of Bologna, 40127 Bologna, Italy; erica.neri4@unibo.it (E.N.); federica.genova@unibo.it (F.G.); 3Center for Reproductive Medicine, Department of Obstetrics and Gynecology, AUSL-IRCCS, 42123 Reggio Emilia, Italy; alessia.nicoli@ausl.re.it (A.N.); mariateresa.villani2@ausl.re.it (M.T.V.); santi.sara@ausl.re.it (S.S.)

**Keywords:** healthcare workers, reproductive medicine, COVID-19 lockdown, mental health, psychological wellbeing, emotions, stress impact, coping

## Abstract

The impact of the COVID-19 pandemic on global healthcare workers’ (HCWs) mental health has been well documented in the last two years; however, little is known regarding HCWs working in specific healthcare fields. During two subsequent periods of national lockdown in Italy (June–July 2020, T1, and November–December 2020, T2), a total sample of 47 HCWs working in a reproductive medicine hospital unit completed an ad hoc questionnaire for assessing emotional reactions to the pandemic, stress symptoms, and ways of coping. Moderate–high levels of anger and sadness were experienced by 65.9% and 68.1% of the HCWs, respectively, while moderate–high levels of anxiety and fear were experienced by 51.1% and 56.8%, respectively. Higher stress symptoms experienced by HCWs were hypervigilance, avoidance of thoughts and memories, and tiredness/low energy. At T2, levels of hypervigilance, irritability, intrusive thoughts, and detachment were higher than at T1, while avoidance of external triggers decreased. Moderate–high levels of anxiety resulted significantly associated with several symptoms of stress: irritability/fearfulness, depression/hopelessness, tiredness/low energy, problems with concentration, and intrusive thoughts. Regarding coping strategies, HCWs tended to adopt more problem-focused coping (e.g., contributing to improving a situation) and this tendency was higher at T2. Overall findings suggest a risk for the persistence of stress symptoms and, therefore, a risk for a chronic course, which might interfere with the global quality of mental health at work and the care provided to patients. Clinical implications highlight the relevance of implementing support programs for this category of HCWs focused on the elaboration of negative emotions and on fostering adaptive coping strategies.

## 1. Introduction

COVID-19 was declared a pandemic by the World Health Organization (WHO) on 11 March 2020. Its rapid spread and high mortality imposed unprecedented and massive changes on society and people’s lives throughout the world [1].

Italy was the first European country to face the COVID-19 outbreak, in a period characterized by a limited knowledge about the virus. At the beginning of the pandemic, the highest incidence was in the northern regions; in Lombardy, Piedmont, Emilia Romagna, and Veneto, more than 10,000 cases of infected individuals were counted (ISS, 12 March 2020). Facing this emergency, the Italian Government implemented extraordinary measures to limit viral transmission, including social distancing, closure of educational and employment facilities, border closure, and lockdown throughout the entire country [2] that persisted until June 2020. Despite a partial improvement in the pandemic condition during the summer months, a sudden increase in coronavirus cases and COVID-related deaths during fall–winter 2020, required the re-activation of severe restrictions.

These exceptional restrictive measures have put an enormous strain on the Italian population [3,4]. In particular, healthcare workers (HCWs) directly involved in dealing this emergency [5] were more exposed to stressors related to the pandemic, reporting high levels of stress, depression, anxiety, somatic symptoms, post-traumatic stress disorders, and sleep disturbances [6,7,8,9,10].

The outbreak control measures affected, more generally, all hospital medical fields, resulting in the formulation and implementation of new guidelines, the reorganization of units and services, the reallocation of healthcare resources to COVID-19 wards, the activation of protocols for containing viral transmission, and the suspension of all non-urgent cares [11]. Although frontline HCWs reported a greater vulnerability to psychological distress than others HCWs [12,13], the impact of the pandemic on HCWs’ mental health has been overall documented, regardless of direct or indirect management of COVID-19 patients.

Many international studies have, in fact, underlined how HCWs experienced extreme work pressure, including work overload, changes in professional tasks and fast adaptations to new health strategies, and facing the risk of being infected or infecting family members on a daily basis. In this scenario, they experienced intense feelings of stress, anxiety, fear, helplessness and loneliness, heightened by the lack of social support due to the lockdowns [14,15,16]. In particular, according to recent systematic umbrella reviews and meta-analyses [1,17], the pooled prevalence rates of mental health symptoms of HCWs in practice, during the COVID-19 pandemic, were about 30% for depression, 30% for anxiety, 30% for post-traumatic and acute stress disorders, and 40% for sleep disorders, with substantial heterogeneity among studies. This variability has been explained by different factors, such as different mortality and infection rates among countries, gender differences, professional category, kind of users, health care setting, working conditions, etc.

It is worth noting that most of the studies included in the abovementioned reviews and meta-analyses were conducted during the first COVID-19 wave (especially from March 2020 to July 2020), while there is a paucity of studies on the impact of the pandemic on HCWs’ mental health during the second wave (November 2020–December 2020). According to the few existing studies conducted during the second wave [18,19,20], more than 40% of HCWs showed mild to moderate anxiety symptoms. Besides, the longitudinal study by Canal-Rivero et al. [21], conducted involving different samples across the first and the second wave, observed a significant improvement in the stress reactions in HCWs, even if the proportion of HCWs who fulfilled the criteria for acute stress disorders (ASDs) did not change over the follow-up period. Also, as suggested by another longitudinal recent study [22] involving different samples across two time points, the severity and prevalence of anxiety, depression, and stress tended to increase at the second peak of the pandemic, compared to the first one. These emerging findings may suggest the existence of a risk for a chronic course of psychological difficulties, which needs further investigations.

Given the severe impact of COVID-19 on HCWs, a line of research has focused on the investigation of their possible coping strategies [23,24,25], suggesting that HCWs with higher levels of anxiety and stress tend to more frequently adopt dysfunctional coping mechanisms (such as avoiding behaviors, self-blaming, rumination), which might worsen the course of the clinical picture.

Considering this risk for long-term consequences on HCWS’ mental health, it could be relevant to explore the impact of the COVID-19 pandemic on HCWs working in specific healthcare contexts, where the patients may require specific care and support. These studies might highlight the amount of risk incurred by different categories of HCWs and might allow the planning of intervention tailored to their specific needs, in order to decrease their vulnerability to psychological problems.

In this context, the field of reproductive medicine was one of the most affected by the unexpected COVID-19 pandemic and by the consequent introduction of control measures. In fact, besides the burden connected to reorganization of the services and tasks, along with the fear of being infected and a heightened lack of social support, HCWs have been faced suspension of all new medically assisted reproduction (MAR) treatment cycles for an indefinite period. HCWs have also experienced a lack of guidelines for dealing with the clinical management of patients and uncertainty about the impact of COVID-19 infection on pregnancy and neonatal outcomes [26,27,28]. In addition, all of this happened to the management and the care of an emotionally and psychologically sensitive category of patients [29,30], that of infertile women and men, who, in response to the COVID-19 emergency, showed an increase in levels of stress, anxiety, and depression that were much higher than the one would normally expect in the infertile population [31,32,33,34].

However, to our knowledge, studies evaluating the psychological influence of the COVID-19 pandemic on the wellbeing of HCWs working with infertile population are lacking. Enhancing this knowledge might shed light on HCWs’ potential vulnerabilities and needs, with relation to the specific characteristics of these hospital units and their patients.

Based on these premises, a staff of psychologists working in a reproductive medicine unit of a public hospital developed an explorative investigation in order to understand the possible influence of the pandemic on stable workers’ psychological wellbeing. A further and more general aim, based on clinical demands, was to better identify the specific needs and the potential risk factors for their mental health, in order to plan possible tailored interventions during the pandemic.

Therefore, the first aim of the study was to investigate the impact of the spread of COVID-19 and of the lockdown periods (associated with the first and second waves) on the mental health of HCWs who continued to work in a center of reproductive medicine; specifically, their range of positive and negative emotions, stress symptoms, and ways of coping were considered, analyzing possible differences between the first and second COVID-19 waves. Secondly, the possible relationship among emotional reactions, stress symptoms and ways of coping were investigated. The goals of the study were mainly exploratory, due to the exceptionality of the event of the COVID-19 pandemic; therefore no specific hypotheses were developed.

## 2. Materials and Methods

This cross-sectional study was realized in Italy in the two different periods of the year 2020, specifically during the first wave period (June–July 2020, Time 1 or T1) and the second one (November–December 2020, Time 2 or T2).

### 2.1. Participants

A total of 47 participants were recruited, whose main socio-demographic characteristics are shown in Table 1.

At T1, June–July 2020, almost all the of HCWs working in the unit completed the questionnaire, 25 out of 27, which represents 92.6%. Twenty-three were women and two were men, mean age 38.5 (SD 7.68). Subjects were recruited at the Center of Reproductive Medicine, Department of Obstetrics and Gynecology, Azienda Unità Sanitaria Locale-IRCCS in Reggio Emilia (Italy), and they were all working permanently in that unit. Socio-demographic characteristics of this sample, called the T1 group, are shown in Table 1.

At T2, November–December 2020, 22 HCWs were still working in the unit and they all completed the questionnaire. The subjects were 19 women and three men, always working in the same place; mean age of the participants was 41.1 (SD 5.75). The main characteristics of this sample, called the T2 group, are presented in Table 1.

### 2.2. Procedure

Subjects were asked for voluntary and anonymous participation in the study by the head of the Center for Reproductive Medicine, with the support of psychologists working in the same unit. The only criteria used to recruit participants was that they had to work permanently as HCW in the Reproductive Medicine Unit (e.g., biologist, nurse).

First, an official email describing the study was sent by the head of the unit and the psychologists to all the HCWs of the Reproductive Medicine Unit. The study was presented as an exploratory survey on emotional experience and psychological wellbeing during the COVID lockdowns; if the HCWs agreed to participate, they could find a printed copy of a questionnaire to complete anonymously in specific rooms of the unit (e.g., clinic). Along with the questionnaire, realized ad hoc by the staff, subjects gave their informed consent. Considering the restricted number of stable HCWs in the unit and the investigation of personal emotional and psychological issues, data were collected and analyzed in an anonymous way to protect respondents’ privacy and the reliability of the answers.

The study was conducted according to the guidelines of the Declaration of Helsinki and ethically reviewed and approved by the head of the Center for Reproductive Medicine.

### 2.3. The Questionnaire

The questionnaire was specifically designed by the psychological staff of the unit, with the aim to address and investigate the specific indicators of mental health and wellbeing of the HCWs working in the unit. It was composed of a series of specific questions inspired by other self-reported questionnaires developed during the COVID-19 emergency, and it was delivered online to investigate the psychological dimensions of wellbeing and adjustment to stress.

Even if the psychologists were aware of the existence of standardized tools for the measurement of variables like stress symptoms and coping, they deliberately chose to develop a specific questionnaire to facilitate the investigation and expression of emotional issues in HCWs. In fact, the psychological staff aimed to avoid the risk of presenting potentially intrusive items (items on somatic and psychopathological dimensions were reduced) and presenting validated tools already known by the HCWs.

The questionnaire was completed in an anonymous form, and it was composed of the following sections:-Section 1: socio-demographic information and changes in work conditions;-Section 2: direct/indirect contact with COVID-19 (three items): this section aimed to collect information on infection and severe complications (or death) due to the virus, experienced personally or in one’s acquaintances;-Section 3: emotional reactions to the pandemic (nine items): the section explored the intensity experienced as a state of a range of positive and negative emotions (e.g., fear, sadness, anxiety, anger) experienced as a state (not as an emotional trait); the answers were rated on a seven-point Likert scale (1 = not at all, 7 = very much);-Section 4: stress symptoms (14 items): this section evaluated the occurrence of the main common signs of stress during the past two weeks (e.g., hypervigilance, avoidance, flashbacks, detachment); the answers were rated on a four-point Likert scale (1 = not at all to 4 = nearly every day);-Section 5: coping strategies (10 items): the section aimed at evaluating the tendency to use specific ways of coping (e.g., avoid conflicts, rely on others), rating the answers on a four-point Likert scale (1 = strongly disagree, 4 = strongly agree).

### 2.4. Statistical Analysis

All statistical analyses were performed both on the total sample and, separately, on the sample at T1 and the sample at T2.

Descriptive analyses were run in order to verify the homogeneity of the samples on socio-demographic and work condition variables (Pearson’s X^2^ Test for categorical variables and Student t-test for continuous variables).

Statistically significant differences in the mean scores of emotional reactions, levels of distress, and ways of coping were tested by the Mann–Whitney U test. Besides, to evaluate the possible influence of emotional states on the characteristics of stress reactions and ways of coping, scores for each emotional state were first categorized into two categories: low intensity, including all subjects with scores ranging between 1 and 3 points, and moderate–high intensity, for the ones with scores of 4–7 points. Categories were then put in relation to the scores of stress symptoms and of ways of coping, respectively, through the Mann–Whitney U test. To control an over risk of type 2 error, all comparisons were defined a priori and no adjustments were made, as recommended by Streiner and Norman [35].

Statistical analyses were performed using IBM SPSS version 20.0 for Windows. A *p* < 0.05 was considered as significant.

## 3. Results

### 3.1. Sample Characteristics: Socio-Demographic and COVID-Related Information

The main characteristics of the total sample and of T1 and T2 groups (sample recruited during the first and second COVID-19 lockdowns, respectively), are shown in Table 1. No group differences were found in relation to gender, age, job, education level and civil status (all had *p* > 0.05).

Besides, most respondents (93.6%) declared no change in their work conditions during the two lockdowns, considering the workplace and work shift (88% and 100% at T1 and T2, respectively).

Most respondents (91.5%) did not contract COVID-19 during both time assessments (92% and 90.9% for T1 and T2 groups, respectively). Nevertheless, in both groups many respondents (87.2%) had acquaintances infected by COVID-19 (76% and 100% for T1 and T2 samples, respectively); of these, almost half died because of the disease (44.7%, of which 36% and 54.5% were T1 and T2 samples, respectively). Frequencies were overall similar between the T1 and T2 groups, excepted for having acquaintances infected by COVID-19 (*X*^2^(1) = 6.053, *p =* 0.014). Subsequent analyses showed that this variable did not significantly influence the dependent variables of the study (emotional reactions, levels of stress, and ways of coping). For this reason, it was not further included in the statistical analyses.

### 3.2. Emotional Reactions to the Pandemic

Mean scores regarding emotional reactions to the pandemic are shown in Table 2.

Considering the total sample, higher emotional reactions seemed to correspond to anger, sadness, and fear. At T1 and at T2, higher emotional reactions were similar, going from the highest, sadness, to anger and then fear. Happiness was the lowest emotional state experienced by participants both at T1 and at T2.

No significant differences emerged in the levels of emotional reactions to the pandemic between T1 and T2 assessments (all had *p* > 0.05; Table 2).

According to the aims of the study, we then calculated two main categorical scores of emotional reactions (low vs. moderate–high).

Figure 1 shows the frequencies of subjects belonging to the categories for every emotional state. Most of the cases of moderate–high scores were for anger and sadness emotions (65.9% and 68.1%, respectively). Moreover, we found that about half of the subjects showed moderate–high levels of anxiety (51.1%) and fear (56.8%). On the contrary, the frequency of subjects with moderate–high levels of disgust, happiness, and surprise was very low (28.9%, 2.2.%, 11.9%, respectively); therefore, these variables were not considered nor included in subsequent analyses.

#### 3.2.1. Stress Symptoms

Levels of stress symptoms are described in Table 3.

In the global sample, higher scores were observed for the following categories of symptoms: hypervigilance, and avoidance of thoughts and memories; lower scores corresponded to hyperreactivity/physical reactions, distressing dreams, and nightmares.

Considering the T1 group, higher stress symptoms were hypervigilance and avoidance of thoughts; at T2, still hypervigilance and avoidance of thoughts showed the highest scores.

When comparing the T1 and T2 groups, mean scores seemed significantly higher at T2 compared to T1 for the following symptoms: hypervigilance (Mann–Whitney U = 365.000, *p* = 0.007), intrusive thoughts (Mann–Whitney U = 360.500, *p* = 0.044), irritability/fearfulness (Mann–Whitney U = 347.000, *p* = 0.041), and obnubilation (Mann–Whitney U = 346.000, *p* = 0.034). Conversely, the symptom “avoidance of external triggers” showed significantly lower mean scores in the T2 group (Mann–Whitney U = 169.000, *p* = 0.032).

All other items did not significantly differ between the two groups (all had *p* > 0.05).

#### 3.2.2. Ways of Coping

Mean scores regarding ways of coping are described in Table 4.

On the global sample, the items “plan the time of the day” and “contribute to improve a situation” got higher scores, while “suffer the situation” got the lowest one. At T1, the highest scores corresponded to the items “plan the time of the day” and “can get help from others” and, for T2, this was “contribute to improve a situation” and “plan the time of the day”.

No statistically significant differences were observed between the two groups, except for the item “contribute to improve a situation” getting a higher value in the T2 group (Mann–Whitney U = 331.500, *p* = 0.014). All other items did not significantly differ between the T1 and T2 groups (all had *p* > 0.05).

### 3.3. Influence of Emotional States on Stress Symptoms

Mann–Whitney U analyses showed a significant influence of anxiety on the scores of stress symptoms (Table 5).

Specifically, most of the items regarding alterations in mood and cognition domain were significantly higher in the moderate–high anxiety group: “irritability/fearfulness” (Mann–Whitney U = 395.000, *p* = 0.002), “diminished interest/pleasure” (Mann–Whitney U = 384.500, *p* = 0.003), “depression and hopelessness” (Mann–Whitney U = 447.500, *p* < 0.0005), and “tiredness/low energy” (Mann–Whitney U = 391.500, *p* = 0.008). Furthermore, significantly higher scores emerged for the subjects with moderate–high anxiety regarding “problems with concentration” (Mann–Whitney U = 393.000, *p* = 0.001) and “intrusive thoughts” (Mann–Whitney U = 401.000, *p* = 0.003).

When anger was considered, the scores of the “avoidance of external triggers” item were significantly lower in the case of the moderate–high anger group (Mann–Whitney U = 127.500, *p* = 0.023).

When sadness was considered, the group with moderate–high sadness showed significantly higher scores for the items “hypervigilance” (Mann–Whitney U = 303.500, *p* = 0.048) and “flashbacks” (Mann–Whitney U = 310.000, *p* = 0.023), but lower scores at “avoidance of external triggers” (Mann–Whitney U = 114.000, *p* = 0.002).

No significant differences emerged between low and moderate–high fear groups for what concerns stress symptoms.

### 3.4. Influence of Emotional States on Ways of Coping

When the possible association between emotional states and ways of coping was explored (Table 6), only a significant effect of anxiety on the item “suffer the situation” emerged (Mann–Whitney U = 354.000, *p* = 0.037): moderately–highly anxious subjects were shown to suffer the situation significantly more compared to the group with low anxiety.

No significant differences emerged between low vs. moderate–high groups according to the emotional states of fear, anger, and sadness.

## 4. Discussion

The aim of the present study was to explore the psychological wellbeing in HCWs working in the field of reproductive medicine, in order to identify possible risk factors for their mental health and wellbeing at work. Specifically, we assessed the psychological impact of the first and second waves of COVID-19 during the two periods of national lockdowns, the most critical moments of the pandemic period. To our knowledge, this was the first study to describe and explore emotional reactions, stress symptoms, and ways of coping among HCWs working in a reproductive medicine unit during the COVID-19 pandemic.

First, regarding to the exploration of the emotional reactions experienced by HCWs during the pandemic, the most intense were anger, sadness, and fear, while the least marked was happiness. The predominance of negative emotional states is in line with other recent studies [36,37,38], particularly with the observation whereby HCWs tend to experience mainly feelings of sadness and anger (ibidem). No significant differences between the first and second waves were observed in the intensity of emotional states; feelings of anger, sadness, and fear were confirmed to be higher in both assessments, showing a slight increase over time. These results seem to prompt that the negative impact of COVID-19 on HCWs emotional wellbeing persists over time, especially in correspondence with both critical moments of the national lockdowns.

Secondly, when the quality and intensity of stress symptoms experienced by HCWs during the pandemic were investigated, several signs of a global burden emerged. Indeed, considering the total sample, the potential symptoms with higher mean scores were represented by hypervigilance and irritability/fearfulness (both regarding the altered arousal domain), avoidance of thoughts and memories (persistent avoidance domain), intrusive thoughts (intrusion symptoms domain), and tiredness/low energy (alterations in mood and cognition). This clinical picture of altered functioning in different domains is coherent with previous recent investigations exploring the impact of the pandemic on trauma-related symptomatology [39,40], whereby mixed samples of frontline and non-frontline HCWs reported higher symptoms of hyperarousal, avoidance, and intrusiveness. These results seem to suggest that the stress experienced by HCWs during the COVID-19 pandemic appears as a persistent perturbation in several areas of mental wellbeing. Because these stress symptoms could be related to a posttraumatic stress disorder, their occurrence is of clinical relevance and should promote the monitoring of trajectories across time.

With regard to the comparison of stress symptoms between the two assessments, during the second wave compared to the first wave, HCWs reported significantly higher symptoms of irritability/fearfulness, hypervigilance (both from the altered arousal domain), intrusive thoughts (intrusion symptoms domain), and detachment/obnubilation (alterations in mood and cognition); they also reported a decrease in the tendency to avoid external triggers of the pandemic. These differences seem to suggest how the prolonged condition of stress (to face the pandemic and the lockdowns) reinforces in HCWs the perception of a status of alarm and alert, both internally and externally, along with an increase in symptoms like intrusion and detachment, which might significantly interfere with their cognitive ability to focus, and the mind clarity needed at work [41,42]. To our knowledge, only one previous study, including both frontline and non-frontline professionals [21], examined changes in stress responses over a six-month period during the pandemic, showing a significant improvement in the hyperarousal dimension over the follow-up period, but no significant differences in the prevalence of acute stress disorder (ASD) between the first and the second assessment. Interestingly, the HCWs also showed a decrease in avoiding external reminders of the pandemic, suggesting a possible habituation and a more active adaptation to the exceptional situation. This finding might be consistent with a recent study realized on a sample of the general Italian population, where during the second wave an increase in a sense of self-efficacy and risk propensity emerged [43].

These results would suggest that the persistence of the pandemic might exacerbate an overall increase in symptoms of maladjustment, with consequent higher risks of developing adjustment disorders (Ads), ASD or posttraumatic stress disorders (PTSD), due to a condition of chronic stress. Therefore, these findings, along with the recent literature on the topic, would strengthen the relevance of implementing longitudinal screening programs to detect higher-risk cases, which would eventually benefit specific forms of support.

Third, with regard to the exploration of the ways of coping with facing the COVID-19 pandemic, the results on the total sample showed that the strategies with higher scores were those involving the ability to solve problems by actively facing them (specifically items: contribute to improve a situation, plan the time of the day), as well as those focused on social support (e.g., can get help from others). These results are in line with previous studies involving mixed samples of frontline and non-frontline professionals, where planning and active coping emerged as the most used coping strategies [23,24,44,45]. These studies also underlined that problem-focused coping resulted as an adaptive strategy and was useful to reduce levels of anxiety [24], depression, and stress [23,46]. We might conclude that the adoption of coping strategies based on problem-focused coping could be particularly relevant for HWCs, because they had to continue to work during the first and second waves, and they had to be particularly focused on an operative approach.

When we considered the comparisons between the groups at T1 and T2, the coping strategy “contribute to improve a situation” was the only one to show a significant improvement over time, suggesting that this way of coping has been particularly salient during the second lockdown. Specifically, given the prolonged pandemic condition, the tendency to use problem-focused coping strategies since the first wave, may have given positive feedback for HCWs in managing daily stresses, and this may have lead them to become progressively more familiar with the up-to-date guidelines for treating MAR patients, become more aware about the risks connected to the impact of COVID-19 infection on pregnancy and neonatal outcomes, and become, globally, more confident and proactive in their working role. This is consistent with the study by Marcolongo et al. [45], where the authors concluded that problem-focused coping strategies allowed, during the second wave, appropriate action to be taken after the management of distress levels. A possible improvement in feelings of self-confidence and self-efficacy is in line with another item that, on the contrary, received the lowest score, especially at T2, that is “suffer the situation” (avoidance domain), suggesting a decrease in the tendency to be passive in contrasting the difficulties, avoiding actively facing the stress.

Taken together, these results suggest that HCWs working in a reproductive medicine unit during the COVID-19 pandemic tended to use quite adaptive coping strategies, which are potentially protective for their mental health problems.

Based on the goals of the study, we also investigated the possible relationship among the levels of negative emotions experienced by the HCWs and their stress and type of coping strategies, respectively.

Regarding the possible relations between emotional states and stress symptoms, the results showed that HCWs experiencing, at that moment, moderate–high levels of anxiety tended to experience more frequent problems of concentration, intrusive thoughts, irritability/fearfulness, depression/hopelessness, tiredness/low energy, and a decrease of interest/pleasure. This result seemed to suggest a role played by the emotion of anxiety on almost all areas of distress, which is typical of the phenomenological manifestation of acute and post-traumatic stress symptomatology [47]. Interestingly, no significant differences regarded the domain of avoidance. Conversely, we found an influence of anger and sadness as, in both cases, subjects with moderate–high levels experienced less frequent avoidance of external triggers. To our knowledge, no other studies assessed the influence of anger and sadness on the levels of distress in HCWs, so further studies would be recommended to better understand and confirm this relationship.

Finally, when fear was considered, the result showed no significant associations with stress signs. Interestingly, despite anxiety and fear theoretically being contiguous constructs, in this study the last one seemed not to be significantly related to stress symptoms. These results might suggest that the exacerbation of stress symptoms in HCWs was mostly related to a state of worry and alertness (typical for anxiety) rather than to the perception of an immediate danger (typical of fear) [47,48]. This is coherent with the stressful condition experienced by HCWs and represented by the spread of the pandemic. While fear usually leads people to a fight or flight action in order to reduce the impact of the threat [49,50,51], within the context of the COVID-19 pandemic these reactions could not be fully expressed because the presence of COVID-19 did not represent a tangible threat to deal with; furthermore, lockdown restrictions reduced the possibility to actively contrast the danger. Therefore, despite fear scores being the highest observed in our sample, their associations with stress appeared limited.

Globally, these results suggest a stronger influence of the emotional state of anxiety on several stress symptoms, and this is coherent with a previous study [36] according to which the prevalence of negative emotional states, characterized by high levels of anxiety (among sadness, irritation, worry, nervousness, fear, and agitation), was strictly related to the severity of stress-related symptoms. Also, our findings are in line with Bassi et al. [52], who studied the predictive role of positive mental health on provisional PTSD diagnoses for health workers during the COVID-19 pandemic in Lombardy: their results showed that languishing emerged as a potential risk factor for PTSD, whereas flourishing proved to be a potential protective factor.

From a clinical perspective, all these findings support the relevance of developing targeted interventions to protect HCWs’ mental health because, as evidenced by the literature [53,54], poor mental health due to the impact of the pandemic might affect the quality of care provided by HCWs, and could lead to long-term psychological difficulties. These considerations are strengthened by the results of an interesting study [37] where mental health professionals (MHPs) reported better overall mental health and lower post-traumatic symptoms than the symptoms observed in other kinds of HCWs during the pandemic. The reasons suggested by the researchers were that MHPs could rely on appropriate strategies of affective and emotional regulation, therefore being able to better handle the stressful impact of the pandemic and to support patients and other health professionals.

Regarding the relations between emotional states and ways of coping, our results, globally, showed no significant associations among the two constructs, suggesting that they seemed not to be strictly related. In particular, these results could be explained considering that the scores of the coping strategies seemed quite similar over time and that the total sample showed a general adoption of functional coping strategies, as represented by a higher use of problem-focused coping. Recent literature has focused on the relationship among psychological disorders (anxiety, depression) and coping strategies [12,23,55,56], finding a significant association between more dysfunctional coping methods and higher symptomatology. To our knowledge, this is the first study giving a first insight on the role played by emotional states on ways of coping during the COVID-19 pandemic, before the psychopathology has already occurred. Interestingly, the only significant relationship regarded anxiety and the coping strategy “suffer the situation”: HCWs with higher levels of anxiety were shown to suffer the situation more frequently than subjects with low anxiety, confirming again the role played by anxiety as a risk factor for mental wellbeing.

This study has several limitations, which need to be mentioned. First, the limited sample size of HCWs of the reproductive medicine unit included in the study might impact on the reliability of results. In fact, this limitation did not allow the consideration of the possible influence on emotions, stress, and coping of variables such as gender, age, and education. Moreover, the small sample size, along with to the multiple testing required by the aims of the study, led us to consider our results as preliminary, and would benefit from further confirmation. However, the size could not be larger as this study was intentionally directed at exploring the level of wellbeing and potential emotional vulnerability in all the professionals working in that specific unit; considering the acceptance rate at T1 and at T2, almost all of them participated in the survey.

Second, this is not a longitudinal study because, even if it concerns the first and second waves of the pandemic, we could not guarantee that respondents at T1 were exactly the same at T2, due to anonymous participation. Therefore, favoring a prudential approach, data were analyzed as a between subjects research design, not as a within subject design. Third, the use of a non-standardized questionnaire for assessing psychological dimensions and coping strategies did not permit statistical analyses with the use of cut-off values to identify the significance at a clinical level; therefore, the results, despite promising, may be read at a descriptive level.

Nevertheless, the findings of this study may help to expand the empirical literature and the understanding of the impact of the COVID-19 pandemic on the emotional and psychological reactions of HCWs working in reproductive medicine wards, and they suggest the usefulness of providing support to prevent the risk of long-term consequences on their mental health.

## 5. Conclusions

Despite the impact of the COVID-19 on HCWs’ dimensions of health being widely investigated since the spread of the pandemic, studies have mainly focused on frontline professionals, while less attention has been paid to specific categories of healthcare workers, such as those working in the field of reproductive medicine. To our knowledge, the present study is the first to examine the reactions of this category of HCWs in terms of emotions, stress symptoms, and ways of coping during the pandemic period.

Globally, the results showed a range of negative emotions and stress symptoms, during both the first and second COVID-19 waves, related to the main domains of potential trauma-related disorders, such as Ads, ASD or PTSD. Besides, negative emotional states appeared to be related with several different stress symptoms.

Based on these findings, the attention paid to HCWs’ emotional states takes shape as a salient issue, especially during the critical conditions caused by the COVID-19 pandemic, as also emerged from the literature [1,17]. Understanding HCWs’ main mental health and wellbeing issues is important in order to provide psychological support for fostering the expression and elaboration of negative emotions; indeed, ad hoc tailored interventions could promote the improvement of appropriate coping and self-regulation strategies, decreasing the risk for a chronic course of stress-related symptoms.

Overall, the present study expands on the recent literature on the topic, with potential research, clinical, and practical implications for the support of HCWs’ mental health and wellbeing under relevant, persistent, and potentially traumatic conditions.

## Figures and Tables

**Figure 1 ijerph-19-05899-f001:**
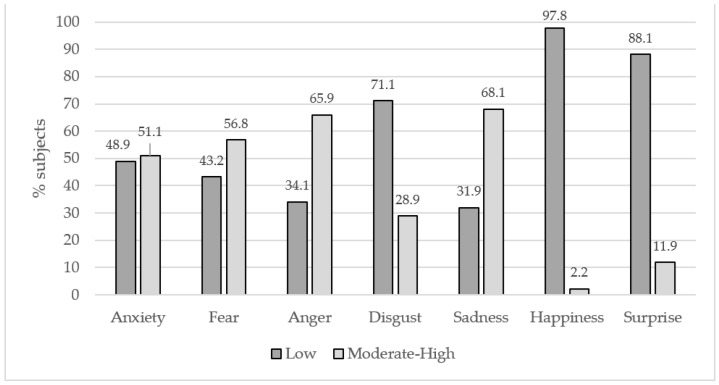
Emotional reactions to pandemic. Values are expressed as frequencies (%).

**Table 1 ijerph-19-05899-t001:** Main characteristics of the sample.

		Total Sample (*N* = 47)	T1 Group (*N* = 25)	T2 Group (*N* = 22)
Age, mean ± SD (range)		39.85 ± 6.76 (27–54)	38.50 ± 7.68 (27–54)	41.07 ± 5.75 (31–50)
Gender (%)	Female	93.2	95.8	90.0
Job (%)	Biologist	19.1	20	18.2
	Nurse	34.0	36	31.8
	Physician	25.5	24	27.3
	Healthcare assistant	12.8	12	13.6
	Other	8.6	8.0	9.1
Education (%)	High school	20.5	27.3	13.6
	Master’s degree	27.3	27.3	27.3
	Specialization	45.5	40.9	50.0
	Other	6.7	4.5	9.1
Civil status (%)	Married/cohabiting	80.8	76.0	86.3
	Single	12.8	16.0	9.2
	Other	6.4	8.0	4.5

**Table 2 ijerph-19-05899-t002:** Emotional reactions to the pandemic.

	Total Sample (*N* = 47)	T1 Group (*N* = 25)	T2 Group (*N* = 22)
Anxiety	2.51 ± 0.80 (1–5)	2.40 ± 0.71 (1–4)	2.64 ± 0.91 (1–5)
Fear	3.82 ± 1.76 (1–7)	3.75 ± 1.80 (1–7)	3.90 ± 1.74 (2–7)
Anger	4.41 ± 2.27 (1–7)	4.33 ± 2.46 (1–7)	4.50 ± 2.10 (1–7)
Disgust	2.74 ± 2.21 (1–7)	2.33 ± 2.06 (1–7)	3.18 ± 2.32 (1–7)
Sadness	4.23 ± 2.02 (1–7)	4.48 ± 1.94 (1–7)	4.77 ± 1.57 (2–7)
Happiness	1.30 ± 0.96 (1–7)	1.25 ± 0.53 (1–3)	1.36 ± 1.29 (1–7)
Surprise	2.07 ± 0.53 (1–7)	1.87 ± 1.33 (1–5)	2.27 ± 1.72 (1–7)

Values are expressed as mean scores ± SD (range).

**Table 3 ijerph-19-05899-t003:** Stress symptoms.

Dimensions	Total Sample (*N* = 47)	T1 Group (*N* = 25)	T2 Group (*N* = 22)
*Altered arousal*			
Irritability and anger	2.18 ± 0.76 (1–4)	2.20 ± 0.56 (1–4)	2.22 ± 0.81 (1–4)
Problems with concentration	1.67 ± 0.60 (1–4)	1.67 ± 0.62 (1–4)	1.67 ± 0.69 (1–4)
Hyperreactivity/physical reactions	1.15 ± 0.56 (1–4)	1.07 ± 0.26 (1–4)	1.11 ± 0.47 (1–4)
Hypervigilance	2.80 ± 1.04 (1–4)	2.60 ± 0.99 (1–4)	3.44 ± 0.78 (1–4) *
*Intrusion symptoms*			
Intrusive thoughts	2.19 ± 0.90 (1–4)	1.93 ± 0.88 (1–4)	2.33 ± 0.77 (1–4) *
Flashbacks	1.64 ± 0.80 (1–4)	1.53 ± 0.80 (1–4)	1.61 ± 0.85 (1–4)
Distressing dreams and nightmares	1.15 ± 0.47 (1–4)	1.00 ± 0.00 (1–4)	1.06 ± 0.24 (1–4)
*Avoidance symptoms*			
Avoidance of external triggers	1.53 ± 0.79 (1–4)	1.53 ± 0.83 (1–4)	1.33 ± 0.77 (1–4) *
Avoidance of thoughts and memories	2.51 ± 0.92 (1–4)	2.47 ± 0.83 (1–4)	2.72 ± 1.02 (1–4)
*Alterations in mood and cognition*			
Irritability/fearfulness	1.74 ± 0.74 (1–4)	1.47 ± 0.52 (1–4)	2.11 ± 0.83 (1–4) *
Detachment/obnubilation	1.54 ± 0.81 (1–4)	1.40 ± 0.83 (1–4)	1.67 ± 0.69 (1–4) *
Diminished interest/pleasure	1.63 ± 0.68 (1–4)	1.67 ± 0.49 (1–4)	1.61 ± 0.78 (1–4)
Depression and hopelessness	1.77 ± 0.73 (1–4)	1.60 ± 063 (1–4)	1.94 ± 0.87 (1–4)
Tiredness/low energy	2.23 ± 0.84 (1–4)	2.33 ± 0.62 (1–4)	2.39 ± 1.09 (1–4)

Values are expressed as mean scores ± SD (range) * *p* < 0.05.

**Table 4 ijerph-19-05899-t004:** Ways of coping.

	Total Sample (N = 47)	T1 Group (N = 25)	T2 Group (N = 22)
*Emotion-focused coping*			
Worry and vent emotions	2.74 ± 0.92 (1–4)	3.00–0.69 (1–4)	2.57–0.98 (1–4)
Try to see the positive side of a situation	3.09 ± 0.93 (1–4)	3.05–0.90 (1–4)	3.00–1.00 (1–4)
*Problem-focused coping*			
Ability to solve problems	3.15 ± 0.66 (1–4)	3.09–0.43 (1–4)	3.19–0.81 (1–4)
Contribute to improve a situation	3.20 ± 0.59 (2–4)	3.00–0.54 (1–4)	3.43–0.60 (1–4) *
Plan the time of the day	3.30 ± 0.83 (1–4)	3.27–0.73 (1–4)	3.29–1.01 (1–4)
*Avoidance-focused coping*			
Avoid conflicts	3.02 ± 0.79 (1–4)	2.91–0.75 (1–4)	3.00–0.83 (1–4)
Suffer the situation	2.20 ± 0.91 (1–4)	2.27–0.88 (1–4)	2.14–0.96 (1–4)
*Support-focused coping*			
Rely on others	2.43 ± 0.85 (1–4)	2.41–0.66 (1–4)	2.43–1.03 (1–4)
Can get help from others	3.17 ± 0.70 (1–4)	3.14–0.64 (1–4)	3.14–0.79 (1–4)
Need of someone’s understanding and support	2.72 ± 0.94 (1–4)	2.64–0.90 (1–4)	2.76–1.00 (1–4)

Values are expressed as mean scores ± SD (range); * *p* < 0.05.

**Table 5 ijerph-19-05899-t005:** Associations among emotional states and stress symptoms.

	Anxiety		Fear		Anger		Sadness	
	Low (*n* = 23)	Moderate–High (*n* = 24)	Low (*n* = 20)	Moderate–High (*n* = 27)	Low (*n* = 16)	Moderate–High (*n* = 31)	Low (*n* = 15)	Moderate–High (*n* = 32)
*Altered arousal*								
Irritability and anger	2.12 ± 0.49	2.29 ± 0.85	2.20 ± 0.67	2.17 ± 0.71	2.00 ± 0.00	2.23 ± 0.81	2.27 ± 0.64	2.17 ± 0.72
Problems with concentration	1.35 ± 0.49	2.00 ± 0.61 *	1.60 ± 0.74	1.67 ± 0.49	1.70 ± 0.48	1.64 ± 0.66	1.64 ± 0.67	1.70 ± 0.64
Hyperreactivity/physical reactions	1.12 ± 0.49	1.06 ± 0.24	1.07 ± 2.26	1.11 ± 0.47	1.00 ± 0.00	1.14 ± 0.47	1.00 ± 0.00	1.13 ± 0.46
Hypervigilance	2.82 ± 1.02	3.24 ± 0.90	2.80 ± 1.01	3.17 ± 0.92	3.00 ± 0.94	3.00 ± 1.02	2.64 ± 0.81	3.22 ± 1.00 *
*Intrusion symptoms*								
Intrusive thoughts	1.76 ± 0.66	2.47 ± 0.87 *	2.07 ± 0.96	2.17 ± 0.79	1.90 ± 0.74	2.23 ± 0.92	1.91 ± 0.94	2.22 ± 0.80
Flashbacks	1.53 ± 0.80	1.59 ± 0.87	1.47 ± 0.83	1.67 ± 0.84	1.40 ± 0.70	1.64 ± 0.90	1.18 ± 0.41	1.74 ± 0.92 *
Distressing dreams and nightmares	1.00 ± 0.00	1.06 ± 0.00	1.00 ± 0.00	1.06 ± 0.24	1.10 ± 0.32	1.00 ± 0.00	1.00 ± 0.00	1.04 ± 0.21
*Persistent avoidance*								
Avoidance of external triggers	1.71 ± 0.99	1.18 ± 0.39	1.60 ± 0.74	1.33 ± 0.84	1.80 ± 0.79	1.32 ± 0.78 *	2.00 ± 0.78	1.17 ± 0.65 *
Avoidance of thoughts and memories	2.65 ± 0.93	2.53 ± 0.94	2.53 ± 0.99	2.67 ± 0.91	2.70 ± 0.82	2.64 ± 0.95	2.45 ± 0.93	2.65 ± 0.94
*Alterations in mood and cognition*								
Irritability/fearfulness	1.41 ± 0.51	2.18 ± 0.81 *	1.67 ± 0.82	1.89 ± 0.76	1.80 ± 0.79	1.73 ± 0.77	1.73 ± 0.79	1.83 ± 0.78
Detachment/obnubilation	1.41 ± 0.62	1.65 ± 0.86	1.67 ± 0.90	1.39 ± 0.61	1.40 ± 0.52	1.50 ± 0.80	1.27 ± 0.47	1.65 ± 0.83
Diminished interest/pleasure	1.41 ± 0.51	1.88 ± 0.70 *	1.67 ± 0.62	1.67 ± 0.69	1.80 ± 0.79	1.64 ± 0.59	1.82 ± 0.60	1.57 ± 0.66
Depression and hopelessness	1.41 ± 0.62	2.18 ± 0.73 *	1.87 ± 0.83	1.72 ± 0.75	1.70 ± 0.82	1.82 ± 0.80	1.82 ± 0.87	1.78 ± 0.74
Tiredness/low energy	1.91 ± 0.79	2.54 ± 0.78 *	2.21 ± 0.98	2.24 ± 0.66	2.07 ± 0.70	2.21 ± 0.82	2.47 ± 0.74	2.13 ± 0.87

*  *p* < 0.05.

**Table 6 ijerph-19-05899-t006:** Associations among emotional states and ways of coping.

	Anxiety		Fear		Anger		Sadness	
	Low (*n* = 23)	Moderate–High (*n* = 24)	Low (*n* = 20)	Moderate–High (*n* = 27)	Low (*n* = 16)	Moderate–High (*n* = 31)	Low (*n* = 15)	Moderate–High (*n* = 32)
*Emotion-focused coping*								
Worry and vent emotions	2.67 ± 0.80	2.91 ± 0.92	2.63 ± 0.83	2.95 ± 0.79	2.79 ± 0.80	2.70 ± 0.87	2.57 ± 0.85	2.90 ± 0.86
Try to see the positive side of a situation	3.19 ± 0.93	2.86 ± 0.94	3.21 ± 0.86	2.91 ± 0.92	3.00 ± 0.87	2.96 ± 0.98	2.93 ± 0.73	3.07 ± 1.03
*Problem-focused coping*								
Ability to solve problems	3.10 ± 0.70	3.18 ± 0.59	3.11 ± 0.57	3.23 ± 0.53	3.07 ± 0.27	3.11 ± 0.75	3.00 ± 0.55	3.21 ± 0.68
Contribute to improve a situation	3.14 ± 0.57	3.27 ± 0.63	3.16 ± 0.50	3.27 ± 0.63	3.21 ± 0.58	3.15 ± 0.60	3.14 ± 0.54	3.24 ± 0.64
Plan the time of the day	3.24 ± 0.94	3.32 ± 0.78	3.26 ± 0.87	3.36 ± 0.73	3.64 ± 0.50	3.04 ± 0.94	3.21 ± 0.98	3.31 ± 0.81
*Avoidance coping*								
Avoid conflicts	2.81 ± 0.87	3.09 ± 0.68	2.84 ± 0.83	3.09 ± 0.61	2.71 ± 0.83	3.00 ± 0.73	2.79 ± 0.80	3.03 ± 0.78
Suffer the situation	1.09 ± 0.83	2.50 ± 0.91 *	2.32 ± 0.88	2.09 ± 0.87	2.21 ± 0.80	2.07 ± 0.87	2.14 ± 0.86	2.24 ± 0.95
*Support-focused coping*								
Rely on others	2.43 ± 0.74	2.41 ± 0.96	2.37 ± 0.76	2.45 ± 0.86	2.50 ± 0.76	2.26 ± 0.81	2.36 ± 0.63	2.45 ± 0.95
Can get help from others	3.24 ± 0.77	3.05 ± 0.65	3.26 ± 0.65	3.09 ± 0.61	3.43 ± 0.51	2.93 ± 0.73	3.07 ± 0.62	3.17 ± 0.76
Need of someone’s understanding and support	2.52 ± 0.93	2.86 ± 0.94	2.79 ± 0.86	2.64 ± 0.95	2.79 ± 0.89	2.56 ± 0.93	2.64 ± 1.01	2.72 ± 0.92

*  *p* < 0.05.

## Data Availability

Data are available upon request due to privacy restrictions.

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
