# Peer review of "Emotions, Stress and Coping among Healthcare Workers in a Reproductive Medicine Unit during the First and Second COVID-19 Lockdowns"

_ijerph, 2022, doi:10.3390/ijerph19105899_

Round 1
Reviewer 1 Report
- An interesting area to be explored. One good thing with this article is, the authors did two-point assessment on similar scenarios (lockdown) to see the changing pattern of mental health issues.
- Are the participants enrolled in T1 and T2 were same?
- Why such small sample was recruited?
- What sampling method was used to recruit the participants?
- What was the selection criteria used to recruit the patients?
- Multiple validated tools were available to measure the psychological domains, by the time the study was conducted. Why the authors developed their own questionnaire and used that?
- The population are heterogeneous. The challenges for different groups of participants will be different. In such a small sample, the findings will be highly biasing.
- Moreover, this article lacks novelty.
Reviewer 2 Report
The authors present the results from a tailored questionnaire study of two samples of staff in two different Italian reproductive medicine units during two waves (June July 2020 and November December 2020) of the covid-19 pandemic. Participation was anonymous so it is not possible to know anything about possible bias due to non-participation but at least it should be possible to get data on how many that received the questionnaire and thus what the overall participation rate was in the two samples.
There is a discussion regarding the cross-sectional nature of the study. However, the authors do not discuss that there were dissimilarities between the samples in that the November-December sample had a higher proportion of specialized and also a higher level of education in general. This could be an important factor in the interpretation of the finding that the subjects participating in wave two had less avoidance and a more active coping pattern (“contribute to improve a situation”). It could be as the authors suggest that this finding may indicate a more active attitude to the problems as a consequence of experiences from the first wave but it could also be due to a difference in advanced care education and experience.
I also note that there is no discussion regarding possible effects of multiple testing. I would not require Bonferroni adjustment but there should be some kind of discussion regarding this problem.
A general problem in the discussion is that the authors (according to my opinion) are overstating their interpretation of the role of anxiety in the observed patterns. There seems to be an assumption that people with “trait anxiety” react more strongly to the situation than others, but it could just as well be that we are looking at “state anxiety” that goes with a lot of other psychological symptoms.
Round 2
Reviewer 1 Report
Revision made is satisfactory
Reviewer 2 Report
Your comments have nuanced the interpretation sufficiently